



# Determination of Natural Frequencies and Mode Shapes of a Wind Turbine Rotor Blades using Timoshenko Beam Elements

Evgueni Stanoev[1], Sudhanva Kusuma Chandrashekhara[1],

[1]Faculty of Mechanical Engineering and Marine Technology, Endowed Chair of Wind Energy Technology, University of Rostock, Rostock, 18059, Germany

*Correspondence to*: PD Dr.-Ing. habil. Evgueni Stanoev (evgueni.stanoev@uni-rostock.de)

**Abstract.** In the simulation of a wind turbine, the lowest eigenmodes of the rotor blades are usually used to describe their elastic deformation in the frame of a multibody system. In this paper, a finite element beam model for the rotor blades based on the transfer matrix method is proposed. Both static and kinetic field matrices for the 3D Timoshenko beam element are derived by numerical integration of the differential equations of motion using RUNGE KUTTA 4th order procedure. The beam reference axis in the general case is at an arbitrary location in the cross section. The inertia term in the motion differential equation is expressed using appropriate shape functions for the Timoshenko beam. The kinetic field matrix is built by numerical integration applied on the approximated inertia term. The beam element stiffness and mass matrices are calculated by simple matrix operations from both field matrices. The system stiffness and mass matrices of the rotor blade model are assembled in the usual finite element manner in a global coordinate system with the accounting for structural twist angle and possibly pre-bending. The program developed for the above calculations and the final solution of the eigenvalue problem is accomplished using MuPAD, a symbolic math toolbox of MATLAB®. The calculated natural frequencies using generic rotor blade data are compared with the results proposed from FAST and ADAMS software.

## 1 Introduction

Vibration of an elastic system refers to a limited reciprocating motion of a particle or an object of the system. Wind turbines operate in an unsteady environment which results in a vibrating response (Manwell, McGowan, & Rogers, 2009). They consist of long slender structures (rotor blades and tower), of which resonant frequencies should be taken into account during the initial design and construction. When the excitation frequency of the vibrating system is near any natural frequency, the undesirable resonant state occurs with large amplitudes, which may lead to damage or even collapse of the wind turbine or its components. The vibration response especially of the rotor blades depends on the stiffness which is a function of the materials used, design and size ( Jureczko, Pawlak, & Mȩżyk, 2005). Therefore, the estimation of natural frequencies in the early design phase plays an important role in avoiding resonance.

The eigenmodes associated to the lowest natural frequencies are employed as shape functions to describe the elastic deformation of the rotor blade beam model in the frame of the usual simulation of the wind turbine as a multi-body system.





Mostly, the first two bending eigenmodes in each (flapwise and edgewise) direction and optionally the two additional torsional eigenmodes are used. The determination of those lowest eigenmodes with sufficient numerical accuracy is the first step for the modal superposition applied to the deformational motion of the rotor blades.

Due to geometrical complexity of the blade cross section profiles and the extended use of composite materials, the exact calculation of natural frequencies in the design process takes a considerable time and financial expense for the 3D modelling of the rotor blade using CAD software. Hence a simplified finite element beam model is necessary. A twisted rotor blade is simplified into a cantilever beam with non-uniform cross section. The structural twist angle is implemented by changing the orientation of the principal axis of the blade cross section along the length of the blade.

In the finite element formulation of beams two linear beam theories are established, the Euler Bernoulli beam model and the

Timoshenko beam model. Although the Euler-Bernoulli beam theory is widely used, the Timoshenko beam theory is considered to be better as it incorporates the effects of transverse shear and the rotational inertia on the dynamic response of the beam (Kaya, 2006). In the classical approach of finite element formulation for the free vibration analysis of beams, the stiffness and mass matrices are derived using interpolation functions derived from second and fourth order Hermite polynomials. The stiffness matrix is derived from the equation (Wu, 2013):

$\underline{K}(e) = \int \underline{B}^T \underline{D_m} \, \underline{B} \, dv$ (1)

where, $\underline{K}(e)$ is the element stiffness matrix, $\underline{B}$ is the strain matrix , $\underline{D_m}$ is the elasticity matrix for the beam. The element mass matrix of the beam is derived using the equation (Wu, 2013):

$\underline{M}(e) = \int \rho \, \underline{a}^T \underline{a} \, dv$ (2)

where, $\underline{M}(e)$ is the element mass matrix, $\rho$ is the mass density, $v$ is the volume and $\underline{a}$ is the matrix of interpolation

functions.

Using the above standard relations and appropriate shape functions for the Euler-Bernoulli beam and Timoshenko beam, the stiffness matrix and consistent mass matrix for the finite beam element can be derived. However, an alternative to this procedure, based on the transfer matrix method for the beam theory, see (Graf & Vassilev, 2006), p.69-88 and (Stanoev, 2007), will be developed in the present article. The element stiffness matrix can be derived on the basis of numerical

integration of the first order ordinary differential equation system for the differential beam element. The associated mass matrix can be developed by numerical integration of the inertia term in the differential equation of motion. The numerical integration results in static and kinetic field matrices, from which the element stiffness and mass matrices can be easily assembled.

In the present article, the above mentioned procedure is used to determine the element stiffness and element mass matrix for

the Timoshenko beam. The interpolation functions used for deriving the mass matrix are based on Hermite polynomials according to (Bazoune & Khulief, 2003). The system stiffness and mass matrices for the rotor blade are assembled in a global coordinate basis in the usual finite element manner. The numerical solution of the associated eigenvalue problem for the system without damping is computed using computer algebra software (in the frame of MATLAB®).





In the section 2 and 3, the proposed method is described in detail and in section 4 the method is applied on a rotor blade structure with 288 DOF. The results for the natural frequencies and the corresponding eigenmodes are compared with the results calculated using FAST and ADAMS software.

## 2 Theoretical framework for 3-D Timoshenko beam

### 2.1 Kinematic relationships

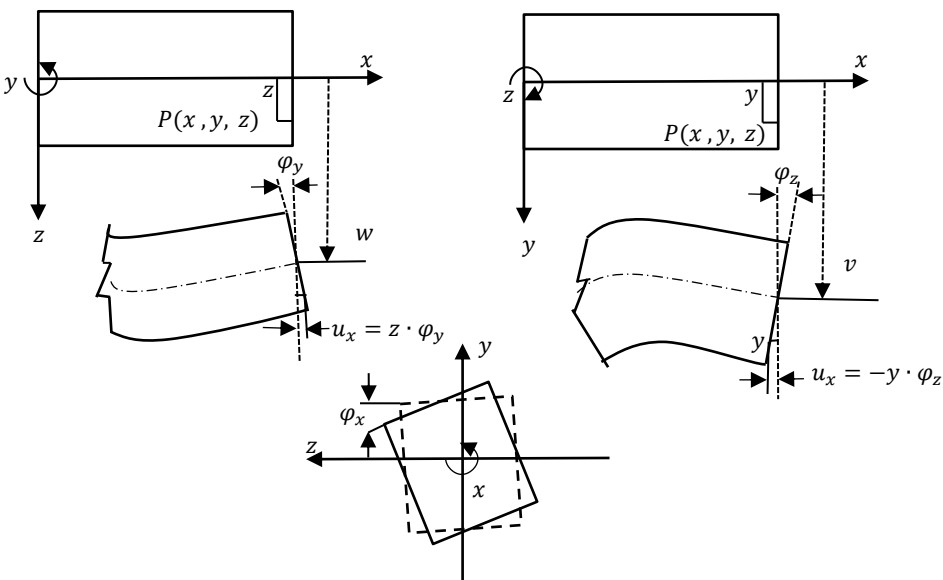

**Figure 1: Deformation at the point $P(x, y, z)$ (Andersen & Nielsen, 2008)**

The general assumptions in the linear beam theory are as follows:

    a) The beam reference (longitudinal) axis is at any arbitrary location of the cross section.

    b) Only stresses that occur are normal stresses $\sigma_x$ and shear stresses $\tau_{xy}, \tau_{xz}$.

c) Cross section planes, which are initially normal to the longitudinal axis, will remain plane after deformation.

The geometrical representation of the deformation state of a beam cross section is shown in the fig. 1. The deformations $u_p(x, y, z)$, $v_p(x, y, z)$, and $w_p(x, y, z)$ at a cross-sectional point $P(x, y, z)$ are determined by three displacement functions $u(x), v(x), w(x)$ and three cross-sectional rotation angles $\varphi_x(x)$, $\varphi_y(x)$ and $\varphi_z(x)$ – all of them are a function of the beam axis coordinate $x$. The differential equation system is derived in accordance to (Stanoev, 2013).

From fig. 1, the displacement vector $\underline{u}_p$ can be expressed at any cross-section point $P(x, y, z)$ as:

$$\underline{u}_p = \begin{bmatrix} u_p(x,y,z) \\ v_p(x,y,z) \\ w_p(x,y,z) \end{bmatrix} = \begin{bmatrix} u(x) - y\varphi_z(x) + z\varphi_y(x) \\ v(x) - z\varphi_x(x) \\ w(x) + y\varphi_x(x) \end{bmatrix} \quad (3)$$





The three components of the strains occurring in the beam can be expressed as the derivatives of the displacement functions $u_p$, $v_p$ and $w_p$. The axial strain $\varepsilon_{xx}$ and the two shear strain components $\gamma_{xz}$ and $\gamma_{xy}$ are given by:

$$\varepsilon_{xx} = \frac{\partial u_p}{\partial x} = u' - y\varphi'_z + z\varphi'_y$$

$$\gamma_{xz} = \frac{\partial u_p}{\partial z} + \frac{\partial w_p}{\partial x} = \underbrace{\varphi_y + w'}_{\gamma_z} + y\varphi'_x \qquad \text{(4a-c)}$$

$$\gamma_{xy} = \frac{\partial u_p}{\partial y} + \frac{\partial v_p}{\partial x} = \underbrace{-\varphi_z + v'}_{\gamma_y} - z\varphi'_x$$

where, $\gamma_z$ and $\gamma_y$ are the constant shear strains which are not neglected in Timoshenko beam theory.

$$\gamma_y = -\varphi_z + v' \qquad \text{(5a)}$$

$$\gamma_z = \varphi_y + w' \qquad \text{(5b)}$$

### 2.2 Principle of virtual work for internal forces

10   The virtual work components for internal forces corresponding to normal stresses and shear stresses are given by:

$$-\delta W_i = \int_x \{\delta u'N + \delta\varphi'_z M_z + \delta\varphi'_y M_y\}\,dx + \int_x \{\delta\gamma_z Q_z\}dx + \int_x \{\delta\gamma_y Q_y\}dx + \int_x \{\delta\varphi'_x M_{TP}\}dx \qquad \text{(6)}$$

Where, $N$ is the axial force, $M_z$ and $M_y$ are bending internal moments, $Q_y$ and $Q_z$ are the corresponding shear forces, $M_{TP}$ is the St. Venant torsional moment.

With the introduction of the constitutive relations of Hooke's material law for the stresses corresponding to the internal

15   forces in (6) and expressing the strains by Eq. (4), (5), the virtual work relationship is reformulated as:

$$
\begin{aligned}
-\delta W_i = \int_x \Big\{ &E\left[\underbrace{\left(\int_A dA\right)}_{A}\cdot u' - \underbrace{\left(\int_A ydA\right)}_{A_y}\cdot \varphi'_z + \underbrace{\left(\int_A zdA\right)}_{A_z}\cdot \varphi'_y\right]\delta u' \\
&-E\left[\underbrace{\left(\int_A ydA\right)}_{A_y}\cdot u' - \underbrace{\left(\int_A y^2 dA\right)}_{A_{yy}}\cdot \varphi'_z + \underbrace{\left(\int_A yzdA\right)}_{A_{yz}}\cdot \varphi'_y\right]\delta\varphi'_z \\
&+E\left[\underbrace{\left(\int_A zdA\right)}_{A_z}\cdot u' - \underbrace{\left(\int_A yzdA\right)}_{A_{yz}}\cdot \varphi'_z + \underbrace{\left(\int_A z^2 dA\right)}_{A_{zz}}\cdot \varphi'_y\right]\delta\varphi'_y \\
&+G\begin{bmatrix}A_{sz}(w'+\varphi_y)\delta w' + A_{sz}(w'+\varphi_y)\delta\varphi_y \\ +A_{sy}(v'-\varphi_z)\delta v' - A_{sy}(v'-\varphi_z)\delta\varphi_z\end{bmatrix} + G\underbrace{\left[\int_A (z^2+y^2)\,dA\right]}_{I_T}\varphi'_x\delta\varphi'_x\Big\}\,dx
\end{aligned}
\qquad \text{(7)}
$$



Here, $A_{sz} = \alpha_{sz} \cdot A$  and  $A_{sy} = \alpha_{sy} \cdot A$ are the shear areas in z and y directions respectively, $\alpha_{sz}$, $\alpha_{sy}$ are the corresponding shear correction coefficients, $A$ is the cross-sectional area, $A_y$ is the static moment with respect to the z axis, $A_z$ is the static moment with respect to the y axis, $A_{yy}$ is the moment of inertia with respect to the z direction, $A_{zz}$ is the moment of inertia with respect to the y direction, $A_{yz}$ is the deviation moment of inertia, $I_T$ is the torsional moment of inertia.

After coefficient comparison in the Eq. (6) and (7) the internal forces corresponding to the normal stresses can be expressed by introducing the cross sectional stiffness matrix $\underline{\underline{EA}}$:

$$\begin{bmatrix} N \\ -M_z \\ M_y \end{bmatrix} = \underbrace{\begin{bmatrix} EA & EA_y & EA_z \\ EA_y & EA_{yy} & EA_{yz} \\ EA_z & EA_{yz} & EA_{zz} \end{bmatrix}}_{\underline{\underline{EA}}} \cdot \begin{bmatrix} u' \\ -\varphi'_z \\ \varphi'_y \end{bmatrix} \qquad \Rightarrow \qquad \begin{bmatrix} u' \\ -\varphi'_z \\ \varphi'_y \end{bmatrix} = \left(\underline{\underline{EA}}\right)^{-1} \begin{bmatrix} N \\ -M_z \\ M_y \end{bmatrix} \tag{8}$$

The shear stress component in Eq. (6) and (7) leads to the following relations:

$$M_x = M_{TP} = GI_T \varphi'_x \tag{9}$$

$$Q_z = GA_{sz}(w' + \varphi_y) \tag{10a}$$

$$Q_y = GA_{sy}(v' - \varphi_z) \tag{10b}$$

For the special case of the cross section coordinate system coinciding with the principal axes, the deformation relationship in Eq. (8) reduces to:

$$\begin{bmatrix} N \\ -M_z \\ M_y \end{bmatrix} = \begin{bmatrix} EA & 0 & 0 \\ 0 & EA_{yy} & 0 \\ 0 & 0 & EA_{zz} \end{bmatrix} \cdot \begin{bmatrix} u' \\ -\varphi'_z \\ \varphi'_y \end{bmatrix} \qquad \Rightarrow \qquad \begin{bmatrix} u' \\ -\varphi'_z \\ \varphi'_y \end{bmatrix} = \left(\underline{\underline{EA}}\right)^{-1} \begin{bmatrix} N \\ -M_z \\ M_y \end{bmatrix} \tag{11}$$

### 2.3    Differential equation system

The virtual work relation in Eq. (7) is rewritten for the case the beam coordinate system coinciding with the principal axis of the cross section – see Eq. (11):

$$-\delta W_i = \int_x \{(EAu')\delta u' + \left(EA_{yy}\varphi'_z\right)\delta\varphi'_z + \left(EA_{zz}\varphi'_y\right)\delta\varphi'_y + (GI_T\varphi'_x)\delta\varphi'_x + GA_{sz}(w' + \varphi_y)(\delta w' + \delta\varphi_y) +$$

$$GA_{sy}(v' - \varphi_z)(\delta v' - \delta\varphi_z)\}dx \tag{12}$$

After partial integration of Eq. (12) the cross section deformation relations and differential equilibrium conditions for the Timoshenko beam element are compiled in a differential equation system of 1st order, see Eq. (13), (14). For the Timoshenko beam with arbitrary beam reference axis at any point on the cross-section, see Eq. (8), the system of differential equations can be expressed in the following form:





$$\frac{d}{dx}\begin{bmatrix} u \\ v \\ w \\ \varphi_x \\ \varphi_y \\ \varphi_z \\ N \\ Q_y \\ Q_z \\ M_x \\ M_y \\ M_z \end{bmatrix} = \begin{bmatrix} 0 & 0 & 0 & 0 & 0 & 0 & S_{11} & 0 & 0 & 0 & S_{13} & -S_{12} \\ 0 & 0 & 0 & 0 & 0 & 1 & 0 & \frac{1}{GA_{sy}} & 0 & 0 & 0 & 0 \\ 0 & 0 & 0 & 0 & -1 & 0 & 0 & 0 & \frac{1}{GA_{sz}} & 0 & 0 & 0 \\ 0 & 0 & 0 & 0 & 0 & 0 & 0 & 0 & 0 & \frac{1}{GI_T} & 0 & 0 \\ 0 & 0 & 0 & 0 & 0 & 0 & S_{31} & 0 & 0 & 0 & S_{33} & -S_{32} \\ 0 & 0 & 0 & 0 & 0 & 0 & -S_{21} & 0 & 0 & 0 & -S_{23} & S_{22} \\ 0 & 0 & 0 & 0 & 0 & 0 & 0 & 0 & 0 & 0 & 0 & 0 \\ 0 & 0 & 0 & 0 & 0 & 0 & 0 & 0 & 0 & 0 & 0 & 0 \\ 0 & 0 & 0 & 0 & 0 & 0 & 0 & 0 & 0 & 0 & 0 & 0 \\ 0 & 0 & 0 & 0 & 0 & 0 & 0 & 0 & 0 & 0 & 0 & 0 \\ 0 & 0 & 0 & 0 & 0 & 0 & 0 & 0 & 1 & 0 & 0 & 0 \\ 0 & 0 & 0 & 0 & 0 & 0 & 0 & -1 & 0 & 0 & 0 & 0 \end{bmatrix} \cdot \begin{bmatrix} u \\ v \\ w \\ \varphi_x \\ \varphi_y \\ \varphi_z \\ N \\ Q_y \\ Q_z \\ M_x \\ M_y \\ M_z \end{bmatrix} + \begin{bmatrix} 0 \\ 0 \\ 0 \\ 0 \\ 0 \\ 0 \\ -p_x \\ -p_y \\ -p_z \\ -m_T \\ -m_y \\ -m_z \end{bmatrix} \quad (13)$$

The differential equation system for the Timoshenko beam with beam reference axis on principal axes can be represented in the following matrix form:

$$\frac{d}{dx}\begin{bmatrix} u \\ v \\ w \\ \varphi_x \\ \varphi_y \\ \varphi_z \\ N \\ Q_y \\ Q_z \\ M_x \\ M_y \\ M_z \end{bmatrix} = \begin{bmatrix} 0 & 0 & 0 & 0 & 0 & 0 & \frac{1}{EA} & 0 & 0 & 0 & 0 & 0 \\ 0 & 0 & 0 & 0 & 0 & 1 & 0 & \frac{1}{GA_{sy}} & 0 & 0 & 0 & 0 \\ 0 & 0 & 0 & 0 & -1 & 0 & 0 & 0 & \frac{1}{GA_{sz}} & 0 & 0 & 0 \\ 0 & 0 & 0 & 0 & 0 & 0 & 0 & 0 & 0 & \frac{1}{GI_T} & 0 & 0 \\ 0 & 0 & 0 & 0 & 0 & 0 & 0 & 0 & 0 & 0 & \frac{1}{EA_{zz}} & 0 \\ 0 & 0 & 0 & 0 & 0 & 0 & 0 & 0 & 0 & 0 & 0 & \frac{1}{EA_{yy}} \\ 0 & 0 & 0 & 0 & 0 & 0 & 0 & 0 & 0 & 0 & 0 & 0 \\ 0 & 0 & 0 & 0 & 0 & 0 & 0 & 0 & 0 & 0 & 0 & 0 \\ 0 & 0 & 0 & 0 & 0 & 0 & 0 & 0 & 0 & 0 & 0 & 0 \\ 0 & 0 & 0 & 0 & 0 & 0 & 0 & 0 & 0 & 0 & 0 & 0 \\ 0 & 0 & 0 & 0 & 0 & 0 & 0 & 0 & 1 & 0 & 0 & 0 \\ 0 & 0 & 0 & 0 & 0 & 0 & 0 & -1 & 0 & 0 & 0 & 0 \end{bmatrix} \cdot \begin{bmatrix} u \\ v \\ w \\ \varphi_x \\ \varphi_y \\ \varphi_z \\ N \\ Q_y \\ Q_z \\ M_x \\ M_y \\ M_z \end{bmatrix} + \begin{bmatrix} 0 \\ 0 \\ 0 \\ 0 \\ 0 \\ 0 \\ -p_x \\ -p_y \\ -p_z \\ -m_T \\ -m_y \\ -m_z \end{bmatrix} \quad (14)$$

The entries $S_{ij}$ in Eq. (13) are determined by inversion of the cross-sectional stiffness matrix in Eq. (8):

$$\begin{bmatrix} u' \\ -\varphi'_z \\ \varphi'_y \end{bmatrix} = \left( \underbrace{\begin{bmatrix} EA & EA_y & EA_z \\ EA_y & EA_{yy} & EA_{yz} \\ EA_z & EA_{yz} & EA_{zz} \end{bmatrix}}_{\underline{\underline{S}}} \right)^{-1} \cdot \begin{bmatrix} N \\ -M_z \\ M_y \end{bmatrix} = \underbrace{\begin{bmatrix} S_{11} & S_{12} & S_{13} \\ S_{21} & S_{22} & S_{23} \\ S_{31} & S_{32} & S_{33} \end{bmatrix}}_{\underline{\underline{S}}} \cdot \begin{bmatrix} N \\ -M_z \\ M_y \end{bmatrix} \quad (15)$$




# 3 Alternative Finite element formulation

In the classical finite element formulation, the beam stiffness matrix and the consistent beam mass matrix are derived by developing an approach for the displacement functions through shape (interpolation) functions, which consists of $2^{nd}$ and $4^{th}$ order Hermite polynomials. In this section, an alternative finite element procedure is presented, based on the numerical

RUNGE KUTTA $4^{th}$ order integration of the differential motion equations. The integration of the static part (the coefficient matrix in Eq. (13) and (14) resp.) leads to the static field matrix, the integration of the inertia terms in the equations of motion Eq. (16) results in a kinetic field matrix. From the last step of the integration, using both field matrices, the element stiffness and resp. the element mass matrices can be calculated by simple matrix operations.

## 3.1 The differential equations of motion

The differential equations of motion for the differential beam element can be written in a matrix form (Stanoev, 2007) as:

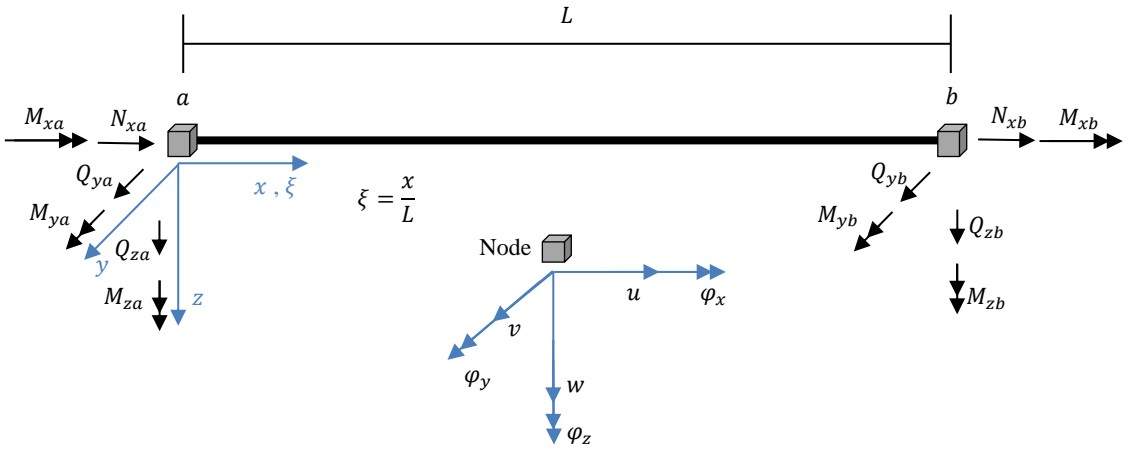

**Figure 2: The finite beam element – cutting forces and nodal DOFs**

$$\begin{bmatrix} \underline{z}_{1,x} \\ \underline{z}_{2,x} \end{bmatrix} = \underbrace{\begin{bmatrix} \underline{A}_{11} & \underline{A}_{12} \\ \underline{A}_{21} & \underline{A}_{22} \end{bmatrix}}_{\underline{A}} \cdot \begin{bmatrix} \underline{z}_1 \\ \underline{z}_2 \end{bmatrix} + \begin{bmatrix} \underline{b}_1 \\ \underline{b}_2 \end{bmatrix} + \begin{bmatrix} \underline{0} \\ \underline{m} \end{bmatrix} \cdot \underline{\ddot{z}}_1 \qquad (16)$$

Where, matrix $\underline{A}$ is the coefficient matrix, see Eq. (13), (14), and vector $\underline{z} = [\underline{z}_1 \quad \underline{z}_2]^T$ is the state vector,

$$\underline{z}_1 = [u(x) \quad v(x) \quad w(x) \quad \varphi_x(x) \quad \varphi_y(x) \quad \varphi_z(x)]^T : \text{vector of the displacement functions} \qquad (16a)$$

$\underline{z}_2 = [N(x) \quad Q_y(x) \quad Q_z(x) \quad M_x(x) \quad M_y(x) \quad M_z(x)]^T :$ vector of cutting (internal) force functions (16b)

The vector $\underline{b} = [\underline{b}_1 \quad \underline{b}_2]^T$ contains the known excitation forces. However, for an eigenvalue problem $\underline{b} = 0$. The coefficient matrix $\underline{A}$ together with the state vector $\underline{z}$ and excitation force $\underline{b}$ constitute the static part of the motion equation.



The kinetic part of the motion equation can be expressed using a matrix of interpolation function and nodal acceleration vector as:

$$
\begin{bmatrix} \underline{0} \\ \underline{m} \end{bmatrix} \cdot \underline{\ddot{z}}_1 = \underbrace{\begin{bmatrix} \underline{0} \\ \underline{m} \end{bmatrix} \cdot \begin{bmatrix} \underline{\Phi}_1(x) & \underline{\Phi}_2(x) \end{bmatrix}}_{\underline{b}_m} \cdot \underbrace{\begin{bmatrix} \underline{\ddot{V}}(a) \\ \underline{\ddot{V}}(b) \end{bmatrix}}_{\underline{\ddot{v}}^{R(e,t)}}
\tag{17}
$$

Where,

$\underline{\ddot{z}}_1 = [\ddot{u} \quad \ddot{v} \quad \ddot{w} \quad \ddot{\varphi}_x \quad \ddot{\varphi}_y \quad \ddot{\varphi}_z]^T$ - Vector of accelerations, $[\underline{\Phi}_1(x) \quad \underline{\Phi}_2(x)] \in R^{(6 \times 12)}$ matrix of interpolation functions (see sec. 3.3), $\underline{\ddot{V}}(a), \underline{\ddot{V}}(b) \in R^{(6 \times 1)}$ vector with nodal accelerations, $\underline{m} \in R^{(6 \times 6)}$ is the inertia matrix of the differential beam element (see Sec.3.2).

### 3.2    The inertia matrix term

The inertia matrix in the Eq. (17) due to distributed mass $\mu(x) \left[\frac{kg}{m}\right]$ implicates eccentrically application of the mass at any location $(y, z)$ in the cross section. The inertia matrix is expressed as (Stanoev, 2013):

$$
\underline{m} = \mu \cdot \begin{bmatrix}
1 & & & & z & -y \\
& 1 & & -z & & \\
& & 1 & y & & \\
& -z & y & y^2 + z^2 + \frac{\Theta_p}{\mu} & & \\
z & & & & z^2 + \frac{\Theta_y}{\mu} & -y \cdot z \\
-y & & & & -y \cdot z & y^2 + \frac{\Theta_z}{\mu}
\end{bmatrix}
\tag{18}
$$

Where, $\Theta_p, \Theta_y, \Theta_z$ in $\left[\frac{kg \, m^4}{m^3}\right]$ are the mass moments of inertia for the cross section:

$$
\Theta_y = \frac{\mu \cdot I_y}{A} = \frac{\mu \cdot A_{zz}}{A}, \quad \Theta_z = \frac{\mu \cdot I_z}{A} = \frac{\mu \cdot A_{yy}}{A}, \quad \Theta_p = \Theta_y + \Theta_z
\tag{19}
$$

### 3.3    Shape functions for Timoshenko beam element

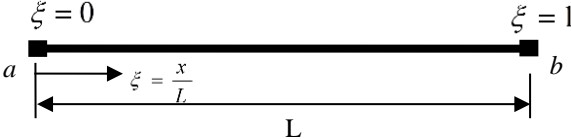

**Figure 3: Definition of dimensionless coordinate (ξ) of a beam element**

The acceleration terms $\underline{\ddot{z}}_1$ in Eq. (17) are expressed using the product of Hemite interpolating polynomials and the nodal acceleration vectors $\underline{\ddot{V}}(a), \underline{\ddot{V}}(b)$.

Shape functions for axial and torsional deformations $u(\xi)$ resp. $\varphi_x(\xi)$ are derived using first order polynomial as:



$$u(\xi) = a_1 + a_2\xi = \underbrace{[1 \quad \xi]}_{\underline{N}_u}\underbrace{\begin{bmatrix} a_1 \\ a_2 \end{bmatrix}}_{\underline{a}} = \underline{N}_u \cdot \underline{a}, \qquad \xi = \frac{x}{L} \tag{20}$$

To express the coefficients $a_j$ in terms of the nodal displacements the following relations $u(\xi = 0) = u_a$,

$u(\xi = 1) = u_b$ resp. and for the torsion $\varphi_x(\xi = 0) = \varphi_{x_a}$, $\varphi_x(\xi = 1) = \varphi_{x_b}$ are applied to Eq. (20):

$$\underbrace{\begin{bmatrix} u_a \\ u_b \end{bmatrix}}_{\underline{u}} = \underbrace{\begin{bmatrix} 1 & 0 \\ 1 & 1 \end{bmatrix}}_{\underline{S}}\begin{bmatrix} a_1 \\ a_2 \end{bmatrix} \quad \rightarrow \quad \begin{bmatrix} a_1 \\ a_2 \end{bmatrix} = \underline{\underline{S}}^{-1} \cdot \underline{u} = \begin{bmatrix} 1 & 0 \\ -1 & 1 \end{bmatrix}\begin{bmatrix} u_a \\ u_b \end{bmatrix} \tag{21}$$

Substituting Eq. (21) in the Eq. (20) results in the shape function for axial deformation

$$u(\xi) = [1 \quad \xi]\underbrace{\begin{bmatrix} 1 & 0 \\ -1 & 1 \end{bmatrix}}_{\underline{G}_u}\underbrace{\begin{bmatrix} u_a \\ u_b \end{bmatrix}}_{\underline{v}_u} = \underline{N}_u \cdot \underline{G}_u \cdot \underline{v}_u = \underbrace{H_{u_1}}_{1-\xi} u_a + \underbrace{H_{u_2}}_{\xi} u_b \tag{22}$$

resp. for torsional deformation $\varphi_x$

$$\varphi_x(\xi) = H_{u_1}\varphi_{x_a} + H_{u_2}\varphi_{x_b} \tag{23}$$

Starting point to derive approximation functions for bending deformation in xz - plane are the relationships (10a), (11) and

the corresponding part of Eq. (14):

$$Q_z = GA_{sz}(w' + \varphi_y) = M_y' = EA_{zz}\varphi_y'' \tag{24}$$

Using the above relation the expression for $w'$ is given by:

$$w' = -\varphi_y + \frac{EA_{zz}}{GA_{sz}}\varphi_y'' = -\varphi_y + \underbrace{\eta_y\frac{L^2}{12}\varphi_y''}_{\gamma_z} \qquad \leftarrow \eta_y = \frac{12EA_{zz}}{GA_{sz}L^2} \tag{25}$$

The translational deformation function $w(\xi)$ is approximated by a cubic polynomial function:

$$w(\xi) = c_0 + c_1\xi + c_2\xi^2 + c_3\xi^3 = \underbrace{[1 \quad \xi \quad \xi^2 \quad \xi^3]}_{N_w}\underbrace{\begin{bmatrix} c_0 \\ c_1 \\ c_2 \\ c_3 \end{bmatrix}}_{c} = \underline{N}_w \cdot \underline{c} \tag{26}$$

Using the constant shear strain relation in Eq. (5b) and Eq. (26) the polynomial expression for constant shear strain can be deduced:

$$\gamma_z = \eta_y\frac{L^2}{12}\varphi_y'' \qquad \text{where,} \quad \varphi_y'' = -w''' = -\frac{6c_3}{L^3} \tag{27}$$

By including Eq. (27) and (26) into Eq. (25) the polynomial expression for $\varphi_y(\xi)$ results in:

$$\varphi_y(\xi) = \frac{1}{L}\left[0 \quad -1 \quad -2\xi \quad -\frac{\eta_y}{2} - 3\xi^2\right]\begin{bmatrix} c_0 \\ c_1 \\ c_2 \\ c_3 \end{bmatrix} = \underline{N}_{\varphi_y} \cdot \underline{c} \tag{28}$$

To determine the coefficients $c_j$ the following boundary conditions are applied:



$$\underline{v}_w = \begin{bmatrix} w_a \\ \varphi_{ya} \\ w_b \\ \varphi_{yb} \end{bmatrix} = \begin{bmatrix} w(\xi = 0) \\ \varphi_y(\xi = 0) \\ w(\xi = 1) \\ \varphi_y(\xi = 1) \end{bmatrix} = \underbrace{\begin{bmatrix} 1 & 0 & 0 & 0 \\ 0 & \frac{-1}{L} & 0 & \frac{-\eta_y}{2L} \\ 1 & 1 & 1 & 1 \\ 0 & \frac{-1}{L} & \frac{-2}{L} & \frac{-\frac{\eta_y}{2}+3}{L} \end{bmatrix}}_{\underline{H}_w} \cdot \begin{bmatrix} c_0 \\ c_1 \\ c_2 \\ c_3 \end{bmatrix} = \underline{H}_w \cdot \underline{c} \tag{29}$$

The inversion of Eq. (29) yields:

$$\underline{c} = \underline{H}_w^{-1} \cdot \underline{v}_w = \frac{1}{1+\eta_y} \cdot \underbrace{\begin{bmatrix} \eta_y+1 & 0 & 0 & 0 \\ -\eta_y & \frac{-L(\eta_y+2)}{2} & \eta_y & \frac{\eta_y}{2}L \\ -3 & \frac{L(\eta_y+4)}{2} & 3 & \frac{-L(\eta_y-2)}{2} \\ 2 & -L & -2 & -L \end{bmatrix}}_{\underline{G}_w} \begin{bmatrix} w_a \\ \varphi_{y_a} \\ w_b \\ \varphi_{y_b} \end{bmatrix} = \underline{G}_w \cdot \underline{v}_w \tag{30}$$

The interpolation functions for $w(x, y, z)$ and $\varphi_y(x, y, z)$, Eq. (26) and (28), can be expressed by employing Eq. (30):

$$w(\xi) = \underbrace{[1 \quad \xi \quad \xi^2 \quad \xi^3]}_{\underline{N}_w} \frac{1}{\eta_y+1} \underbrace{\begin{bmatrix} \eta_y+1 & 0 & 0 & 0 \\ -\eta_y & \frac{-L(\eta_y+2)}{2} & \eta_y & \frac{\eta_y}{2}L \\ -3 & \frac{L(\eta_y+4)}{2} & 3 & \frac{-L(\eta_y-2)}{2} \\ 2 & -L & -2 & -L \end{bmatrix}}_{\underline{G}_w} \underbrace{\begin{bmatrix} w_a \\ \varphi_{ya} \\ w_b \\ \varphi_{yb} \end{bmatrix}}_{\underline{v}_w}$$

$$= H_{w_1} w_a + H_{w_2} \varphi_{y_a} + H_{w_3} w_b + H_{w_4} \varphi_{y_b} \tag{31}$$

Where, the product of both matrices $\underline{N}_w$ and $\underline{G}_w$ is introduced as functions $H_{w_j}$ ($j = 1, .., 4$).

$$\varphi_y(\xi) = \underbrace{\frac{1}{L}\left[0 \quad -1 \quad -2\xi \quad -\frac{\eta_y}{2} - 3\,\xi^2\right]}_{N_{\varphi y}} \frac{1}{\eta_y+1} \begin{bmatrix} \eta_y+1 & 0 & 0 & 0 \\ -\eta_y & \frac{-L(\eta_y+2)}{2} & \eta_y & \frac{\eta_y}{2}L \\ -3 & \frac{L(\eta_y+4)}{2} & 3 & \frac{-L(\eta_y-2)}{2} \\ 2 & -L & -2 & -L \end{bmatrix} \cdot \begin{bmatrix} w_a \\ \varphi_{y_a} \\ w_b \\ \varphi_{y_b} \end{bmatrix}$$

$$= H_{\varphi_{y_1}} w_a + H_{\varphi_{y_2}} \varphi_{y_a} + H_{\varphi_{y_3}} w_b + H_{\varphi_{y_4}} \varphi_{y_b} \tag{32}$$

In Eq. (32), the functions $H_{\varphi_{y_j}}$ ($j = 1, .., 4$) are introduced in an analogous manner.

Similar method is used in determining the approximation functions $v(\xi)$ and $\varphi_z(\xi)$ for bending deformation in xy – plane.
Starting with relations (10b), (11) and (14) to obtain:

$$Q_y = GA_{sy}(v' - \varphi_z) = -M_z' = -EI_z\varphi_z'' \tag{33}$$

$$v' = \varphi_z - \frac{EI_z}{GA_{sy}}\varphi_z'' = \varphi_z + \eta_z \frac{L^2}{12}\varphi_z'' \qquad \leftarrow \quad \eta_z = \frac{12EI_z}{GA_{sy}L^2}, \tag{34}$$

The approximations analogous to Eq. (31) and (32) can be derived:

$$v(\xi) = H_{v_1} v_a + H_{v_2} \varphi_{z_a} + H_{v_3} v_b + H_{v_4} \varphi_{z_b} \tag{35}$$





$$\varphi_z(\xi) = H_{\varphi_{z_1}} v_a + H_{\varphi_{z_2}} \varphi_{za} + H_{\varphi_{z_3}} v_b + H_{\varphi_{z_4}} \varphi_{zb} \tag{36}$$

The $H_{*j}$ –functions developed in Eq. (31), (32) and Eq. (35), (36) are "static" shape functions for the Timoshenko beam. Supposing dependence of time only for the nodal displacement vectors $\underline{V}(a)$, $\underline{V}(b)$, the matrix of the interpolation functions $\left[\underline{\underline{\Phi}}_1(x) \quad \underline{\underline{\Phi}}_2(x)\right]$ in the inertia term Eq. (17) can be developed by using Eq. (31), (32), (35), (36) (Kusuma Chandrashekhara, 2018):

$$\left[\underline{\underline{\Phi}}_1(x) \quad \underline{\underline{\Phi}}_2(x)\right] = \begin{bmatrix} H_{u_1} & & & & H_{u_2} & & & \\ & H_{v_1} & & & H_{v_2} & H_{v_3} & & H_{v_4} \\ & & H_{w_1} & H_{w_2} & & & H_{w_3} & H_{w_4} \\ & & & H_{u_1} & & & H_{u_2} & \\ & & H_{\varphi_{y_1}} & H_{\varphi_{y_2}} & & & H_{\varphi_{y_3}} & H_{\varphi_{y_4}} \\ & H_{\varphi_{z_1}} & & & H_{\varphi_{z_2}} & H_{\varphi_{z_3}} & & H_{\varphi_{z_4}} \end{bmatrix} \tag{37}$$

### 3.4 Numerical Integration

The special form of the numerical RUNGE-KUTTA 4[th] order integration method applied here is described in detail in (Müller & Wolf, 1975) and (Schenk, 2012). The integration operator is applied to the equations of motion in Eq. (16), i.e.

$$\begin{bmatrix} \underline{z}_{1,x} \\ \underline{z}_{2,x} \end{bmatrix} = \underline{\underline{A}} \cdot \begin{bmatrix} \underline{z}_1 \\ \underline{z}_2 \end{bmatrix} + \begin{bmatrix} \underline{b}_1 \\ \underline{b}_2 \end{bmatrix} + \underline{\underline{b}}_m \cdot \underline{\ddot{V}}^R(e, t) \tag{38}$$

In order to gain sufficient numerical precision the beam axis needs to be divided into at least 20 integration intervals. The integration operator transfers the known state vector at beginning of the integration interval to the end of the interval. The integration procedure starts with state vector at the first node $a$, i.e. at location $(x = 0)$:

$$\underbrace{\begin{bmatrix} \underline{z}_1(x = 0) \\ \underline{z}_2(x = 0) \\ 1 \end{bmatrix}}_{\underline{z}(a)} = \begin{bmatrix} 1 & & \\ & \cdots & \\ & & 1 \end{bmatrix} \underline{z}(a) \tag{39}$$

The integration operator is applied subsequently to the evaluated coefficient matrix $\underline{\underline{A}}$ in each interval by excluding the initial state vector $\underline{z}(a)$. The result are static field matrices $\underline{\underline{F}}(x, a)$, multiplicative linked to $\underline{z}(a)$, see Eq.(40). So each $\underline{\underline{F}}(x, a)$-matrix "transfers" the state vector at location $(x = 0)$ to the end $x$ of the considered integration interval (transfer matrix method). In the frame of the integration procedure the actual field matrix $\underline{\underline{F}}(x, a)$ serves column wise as initial vector for the next interval, the components of the state vector $\underline{z}(a)$ remains excluded. The beam "load" vector $\begin{bmatrix} \underline{b}_1 \\ \underline{b}_2 \end{bmatrix}$, Eq. (38), evaluated in the actual interval, yields after integration the $\begin{bmatrix} \underline{\beta}_1 \\ \underline{\beta}_2 \\ 1 \end{bmatrix}$-column in the $\underline{\underline{F}}(x, a)$-matrix – Eq. (40).



The numerical integration of the inertia term $\underline{b}_m$ in Eq. (38) is done column wise analogously to the "load" vector, by excluding the nodal accelerations $\underline{\ddot{V}}^R(e,t)$ – the result are kinetic field matrices $\underline{\underline{H}}(x,a)$ at the end of each integration interval (at location $x$), see Eq.(40):

$$\underbrace{\begin{bmatrix} \underline{z}_1(x) \\ \underline{z}_2(x) \\ 1 \end{bmatrix}}_{\underline{z}(x)} = \underbrace{\begin{bmatrix} \underline{\underline{\alpha}}_{11} & \underline{\underline{\alpha}}_{12} & \underline{\beta}_1 \\ \underline{\underline{\alpha}}_{21} & \underline{\underline{\alpha}}_{22} & \underline{\beta}_2 \\ \underline{0} & \underline{0} & 1 \end{bmatrix}}_{\underline{\underline{F}}(x,a)} \cdot \underbrace{\begin{bmatrix} \underline{z}_1(a) \\ \underline{z}_2(a) \\ 1 \end{bmatrix}}_{\underline{z}(a)} + \underbrace{\begin{bmatrix} \underline{\underline{H}}_{11} & \underline{\underline{H}}_{12} & 0 \\ \underline{\underline{H}}_{21} & \underline{\underline{H}}_{22} & 0 \\ \underline{0} & \underline{0} & 1 \end{bmatrix}}_{\underline{\underline{H}}(x,a)} \cdot \underbrace{\begin{bmatrix} \underline{\ddot{V}}(a) \\ \underline{\ddot{V}}(b) \\ 0 \end{bmatrix}}_{\underline{\ddot{V}}^R(e)} \qquad (40)$$

This type of numerical integration allows (slightly) varying values of the coefficients of the $\underline{\underline{A}}$-matrix, the $\underline{b}_m$-inertia term and of the $\underline{b}$-vector along the beam axis – i.e. all stiffness, mass and external load values of the beam element may vary. After the last integration step at the second node $b$, at location $(x=L)$, static $\underline{\underline{L}}(e)$ and kinetic $\underline{\underline{H}}(e)$ field matrices are obtained:

$$\underbrace{\begin{bmatrix} \underline{z}_1(b) \\ \underline{z}_2(b) \\ 1 \end{bmatrix}}_{\begin{bmatrix}\underline{V}(b) \\ \underline{S}(b) \\ 1\end{bmatrix}} = \underbrace{\begin{bmatrix} \underline{\underline{L}}_{11} & \underline{\underline{L}}_{12} & \underline{f}_1 \\ \underline{\underline{L}}_{21} & \underline{\underline{L}}_{22} & \underline{f}_2 \\ \underline{0} & \underline{0} & 1 \end{bmatrix}}_{\underline{\underline{L}}(e)} \cdot \underbrace{\begin{bmatrix} \underline{z}_1(a) \\ \underline{z}_2(a) \\ 1 \end{bmatrix}}_{\begin{bmatrix}\underline{V}(a) \\ \underline{S}(a) \\ 1\end{bmatrix}} + \underbrace{\begin{bmatrix} \underline{\underline{H}}_{11} & \underline{\underline{H}}_{12} & 0 \\ \underline{\underline{H}}_{21} & \underline{\underline{H}}_{22} & 0 \\ \underline{0} & \underline{0} & 1 \end{bmatrix}}_{\underline{\underline{H}}(e)} \cdot \begin{bmatrix} \underline{\ddot{V}}(a) \\ \underline{\ddot{V}}(b) \\ 0 \end{bmatrix} \qquad (41)$$

According to Eq. (13) and (14) the state variable $\underline{z}_1$ represents 6-component displacement vectors for $\underline{V}(a)$ and $\underline{V}(b)$, respectively, and the state variable $\underline{z}_2$ represents 6-component cutting forces vectors $\underline{S}(a)$ and $\underline{S}(b)$, at locations $(x=0)$ and $(x=L)$ respectively.

### 3.5    The element stiffness and mass matrices

By solving the matrix in Eq. (41) for $\underline{S}(a)$ and $\underline{S}(b)$, and accounting for the definitions of cutting forces in finite element beam formulation, see fig. 2,

$$\underline{F}(a) = -\underline{S}(a), \quad \underline{F}(b) = \underline{S}(b), \qquad (42)$$

one can derive the element stiffness matrix $\underline{\underline{K}}(e)$, element mass matrix $\underline{\underline{M}}(e)$ and element forces and moments $\underline{F}^0$ using simple matrix operations as shown in Eq. (43). Then the beam element relationships for the cutting forces at two nodes can be formulated as:

$$\begin{bmatrix} \underline{F}(a) \\ \underline{F}(b) \end{bmatrix} = \begin{bmatrix} \underline{F}^0(a) \\ \underline{F}^0(b) \end{bmatrix} + \underbrace{\begin{bmatrix} \underline{\underline{K}}_{aa} & \underline{\underline{K}}_{ab} \\ \underline{\underline{K}}_{ba} & \underline{\underline{K}}_{bb} \end{bmatrix}}_{\underline{\underline{K}}(e)} \cdot \begin{bmatrix} \underline{V}(a) \\ \underline{V}(b) \end{bmatrix} + \underbrace{\begin{bmatrix} \underline{\underline{M}}_{aa} & \underline{\underline{M}}_{ab} \\ \underline{\underline{M}}_{ba} & \underline{\underline{M}}_{bb} \end{bmatrix}}_{\underline{\underline{M}}(e)} \cdot \begin{bmatrix} \underline{\ddot{V}}(a) \\ \underline{\ddot{V}}(b) \end{bmatrix} = \underbrace{\begin{bmatrix} \underline{\underline{L}}_{12}^{-1} \cdot \underline{f}_1 \\ \underline{f}_2 - \underline{\underline{L}}_{22} \cdot \underline{\underline{L}}_{12}^{-1} \cdot \underline{f}_1 \end{bmatrix}}_{\underline{F}^0} +$$



$$\underbrace{\begin{bmatrix} \underline{L}_{12}^{-1} \cdot \underline{L}_{11} & -\underline{L}_{12}^{-1} \\ \underline{L}_{21} - \underline{L}_{22} \cdot \underline{L}_{12}^{-1} \cdot \underline{L}_{11} & \underline{L}_{22} \cdot \underline{L}_{12}^{-1} \end{bmatrix}}_{\underline{K}^{(e)}} \cdot \begin{bmatrix} \underline{V}(a) \\ \underline{V}(b) \end{bmatrix} + \underbrace{\begin{bmatrix} \underline{L}_{12}^{-1} \cdot \underline{H}_{11} & -\underline{L}_{12}^{-1} \cdot \underline{H}_{12} \\ \underline{H}_{21} - \underline{L}_{22} \cdot \underline{L}_{12}^{-1} \cdot \underline{H}_{11} & \underline{H}_{22} - \underline{L}_{22} \cdot \underline{L}_{12}^{-1} \cdot \underline{H}_{12} \end{bmatrix}}_{\underline{M}^{(e)}} \cdot \begin{bmatrix} \underline{\ddot{V}}(a) \\ \underline{\ddot{V}}(b) \end{bmatrix} \tag{43}$$

### 3.6 Single masses at eccentric positions

The numerical integration according to RUNGE-KUTTA, described in Sec. 3.4, offers the possibility to include single load or mass quantities within a beam element. Single eccentric masses can be taken into account at the integration interval boundaries. In the local coordinate system of the beam element, at a general position vector $\underline{x}_{AE} = [y_E \quad z_E]^T$ , the eccentric mass and the vector representation of dynamic equilibrium, Eq.(44a-b), is as shown in fig 4. The beam reference axis is at point $A$, and vector $\underline{\ddot{V}}_E$ represents the acceleration vector at the point of application $(x, \underline{x}_{AE})$. With the help of the dynamic equilibrium conditions Eq.(44a-b), additional inertia forces and moments due to eccentric mass can be determined. (Li, 2015)

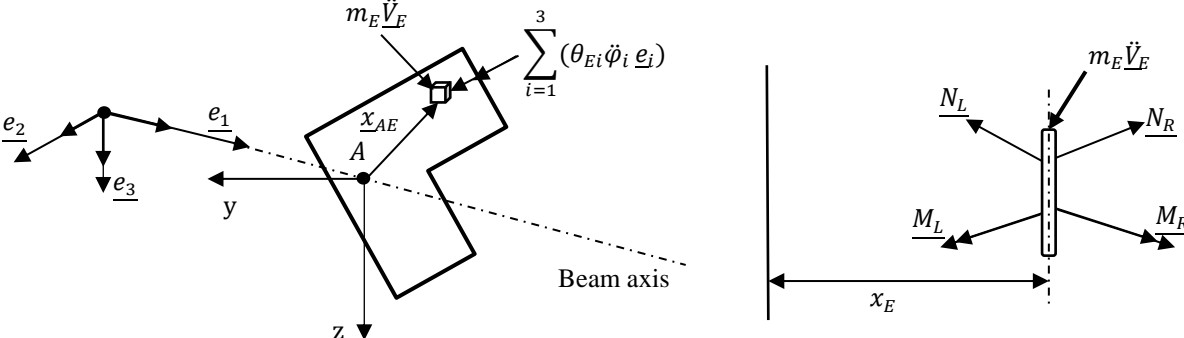

Figure 4: Eccentrically applied mass $m_E$ at the point $\underline{x}_{AE}$ of the beam in the 3D case

$$-\underline{N}_L + \underline{N}_R = m_E \underline{\ddot{V}}_E \tag{44a}$$

$$\underline{M}_L + \underline{M}_R = \sum_{i=1}^{3}(\Theta_{Ei} \, \ddot{\varphi}_i \, \underline{e}_i) + \left(\underline{x}_{AE} \times m_E \underline{\ddot{V}}_E\right) \tag{44b}$$

where, $\underline{N}_l$, $\underline{N}_R$ resp. $\underline{M}_L$ and $\underline{M}_R$ are cutting force resp. moment vectors on left and right side in differential proximity to the point at location $x$ (see fig. 4), $m_E$ is the eccentric single mass, $\Theta_{Ei}$ are the mass moments of inertia of the mass and $\ddot{\varphi}_i$ are the angular accelerations at location $x$.

The additional inertia matrix $\underline{M}_E(m_E, \Theta_{Ei}, x, \underline{x}_{AE})$, derived from Eq.(44a-b), is analogous to the inertia matrix due to distributed mass in the Eq. (18). During the numerical integration within the beam element, see Sec. 3.4, an additional



eccentric inertia term has to be added to the kinetic field matrix at the end $x$ (the point of application of eccentric mass) of the corresponding integration interval – see Eq. (40) and (45).

$$\underline{z}(x) = \begin{bmatrix} \underline{\alpha}_{11} & \underline{\alpha}_{12} & \underline{\beta}_1 \\ \underline{\alpha}_{21} & \underline{\alpha}_{22} & \underline{\beta}_2 \\ \underline{0} & \underline{0} & 1 \end{bmatrix} \cdot \begin{bmatrix} \underline{z}_1(a) \\ \underline{z}_2(a) \\ 1 \end{bmatrix} + \left[ \begin{bmatrix} \underline{H}_{11} & \underline{H}_{12} & \underline{0} \\ \underline{H}_{21} & \underline{H}_{22} & \underline{0} \\ \underline{0} & \underline{0} & 1 \end{bmatrix} + \begin{bmatrix} \underline{0} & \underline{0} & \underline{0} \\ M_E\Phi_1 & M_E\Phi_2 & \underline{0} \\ \underline{0} & \underline{0} & 1 \end{bmatrix} \right] \cdot \begin{bmatrix} \ddot{\underline{V}}(a) \\ \ddot{\underline{V}}(b) \\ 0 \end{bmatrix} \quad (45)$$

Single masses will usually not appear in a rotor blade model, but the same finite element may be used for modelling of wind turbine towers. In this case single masses within a finite beam element could represent bolted ring flange connections or the mass of any equipment like lifts etc.

## 4    The eigenvalue problem

The system matrices for a rotor blade beam model are assembled in the usual finite element manner employing the developed element matrices $\underline{K}(e)$ and $\underline{M}(e)$ from Eq. (43). In the case of free damped oscillation, the linear homogenous differential equations of motion are given by:

$$\underline{M}\,\ddot{\underline{q}}(t) + \underline{D}\,\dot{\underline{q}}(t) + \underline{K}\,\underline{q}(t) = \underline{0} \quad (46)$$

where, $\underline{M} \in R^{(n \times n)}$ is the system mass matrix, $\underline{K} \in R^{(n \times n)}$ is the system stiffness matrix, $\underline{D} \in R^{(n \times n)}$ is the system damping matrix and $\underline{q}(t) \in R^{(n \times 1)}$ is the nodal displacement vector. The system matrices are symmetric and positive definite for finite element structures. For a free undamped system, the matrix for the equation of motion is reduced to:

$$\underline{M}\,\ddot{\underline{q}}(t) + \underline{K}\,\underline{q}(t) = \underline{0} \quad (47)$$

By introducing the following solution approach which is given by:

$$\underline{q}(t) = \hat{\underline{q}}\,e^{i\omega_0 t}, \quad \ddot{\underline{q}}(t) = \hat{\underline{q}}\,(i\omega_0)^2 e^{i\omega_0 t} \quad (48)$$

into the equation of motion Eq. (47) the eigenvalue problem is obtained:

$$\left(\underline{M}^{-1}\underline{K} - \omega_{0k}{}^2\underline{I}\right)\hat{\underline{q}}_k = \underline{0}\ , \quad (49)$$

Where, $\underline{I}$ is a unity matrix. The condition for non-trivial solution for Eq. (49) is given by

$$p(\omega_{0k}{}^2) = det\left(\underline{M}^{-1}\underline{K} - \omega_{0k}{}^2\underline{I}\right) = 0 \quad (50)$$

The n-grade characteristic polynomial $p(\omega_{0k}{}^2)$ has n real solutions $\omega_{0k}, (k = 1, \dots, n)$ (eigenfrequencies) and associated n eigenvectors $\hat{\underline{q}}_k$, calculated from Eq. (49). For real life tasks the solution is usually done by use of eigensolver software.





## 5   Numerical example

The programing code for the procedure described above as for the graphic plots shown below was written in MuPAD, a symbolic math toolbox of MATLAB®, see (Kusuma Chandrashekhara, 2018). The code was verified using realistic data for a wind turbine rotor blade. The blade structural data belongs to a 5 MW reference wind turbine designed for offshore
development (Jonkman, Butterfield, Musial, & Scott, 2009). The blade is of length 63 m divided into 48 beam elements. The blade structural data consists of distributed mass ($m_L$), blade extensional stiffness ($EA$), flapwise stiffness ($EA_{zz}$), edgewise stiffness ($EA_{zz}$), torsional stiffness ($GI_t$), flapwise mass moment of inertia ($\Theta_y$), edgewise mass moment of inertia ($\Theta_z$). For lack of any shear stiffness data in (Jonkman, Butterfield, Musial, & Scott, 2009) the values of ($GA_{sz}$) and ($GA_{sy}$) - the edgewise resp. flapwise shear stiffness – are estimated as 10 % resp. 20 % of extensional stiffness ($EA$) . The values of the
above mentioned stiffness and mass moment of inertias are specified at span wise locations along the blade pitch axis and about the principal axes of each cross section as oriented by a twist angle (γ) defined in the input data. The twist angle is incorporated by using the rotational transformation of each local element stiffness resp. mass matrices (obtained after numerical integration) into the global coordinate system. The results of first three (flapwise and edgewise) eigenfrequencies calculated using Timoshenko beam model, see (Kusuma Chandrashekhara, 2018), are compared with the proposed results
from FAST and ADAMS (Jonkman, Butterfield, Musial, & Scott, 2009). The results are as shown in the table below:

| Description | FAST [$Hz$] | ADAMS [$Hz$] | $f_{\text{Timosh}}$ [$Hz$] | Percentage deviation [%] | |
| --- | --- | --- | --- | --- | --- |
| | | | | FAST and Timoshenko | ADAMS and Timoshenko |
| 1st blade Asymmetric Flapwise Yaw | 0.6664 | 0.6296 | | 0.60 | 6.48 |
| 1st Asymmetric Flapwise Pitch | 0.6675 | 0.6686 | **0.6704** | 0.43 | 0.27 |
| 1st Blade Collective Flap | 0.6993 | 0.7019 | | 4.13 | 4.49 |
| 1st Blade Asymmetric Edgewise Pitch | 1.0793 | 1.0740 | | 1.53 | 2.03 |
| 1st Blade Asymmetric Edgewise Yaw | 1.0898 | 1.0877 | **1.0958** | 0.56 | 0.74 |
| 2nd Blade Asymmetric Flapwise Yaw | 1.9337 | 1.6507 | | 1.78 | 15.05 |
| 2nd Blade Asymmetric Flapwise Pitch | 1.9223 | 1.8558 | **1.8992** | 1.20 | 2.34 |
| 2nd Blade Collective Flap | 2.0205 | 1.9601 | | 6.39 | 3.11 |

**Table 1: First three calculated (bolded values) flapwise and edgewise eigenfrequencies**





The mode shapes and the corresponding eigenfrequencies for the first flapwise and edgewise eigenmodes as well for two
torsional eigenmodes are as shown below:

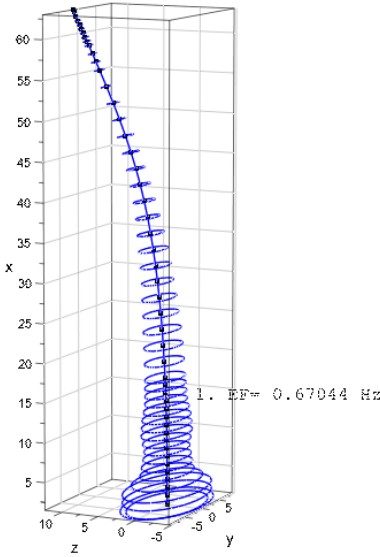

**Figure 5: First flapwise eigenmode**

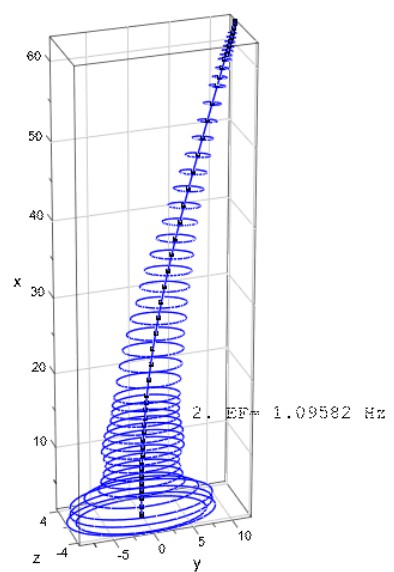

**Figure 6: First edgewise eigenmode**

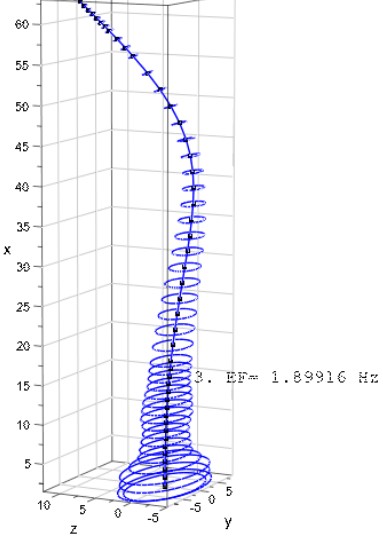

**Figure 7: Second flapwise eigenmode**

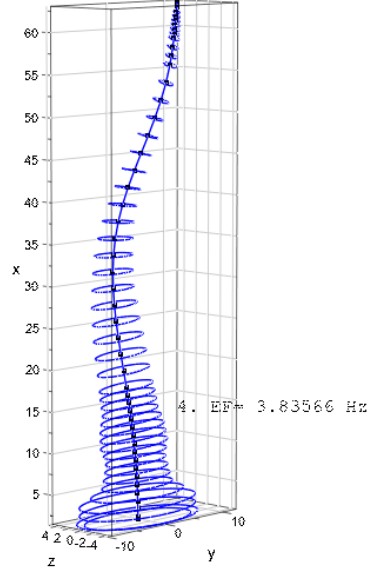

**Figure 8: mixed flap/edgewise eigenmode**



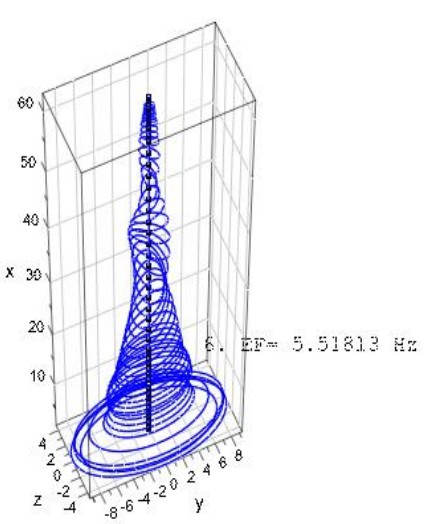

**Figure 9: First torsional eigenmode**          **Figure 10: Third flapwise eigenmode**

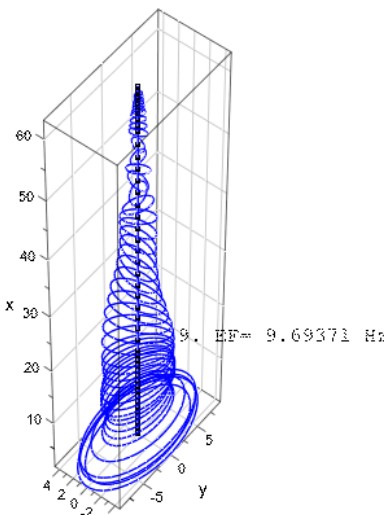

**Figure 11: Second torsional eigenmode**




## 6     Conclusion and Outlook

The proposed Timoshenko beam element in 3D description has been developed on the basis of the transfer matrix method. Both static and kinetic field matrices for the beam element are derived by applying in a special way a RUNGE KUTTA 4$^{th}$ order numerical integration procedure on the differential equations of motion. Appropriate shape functions for the

Timoshenko beam have been used to approximate the inertia forces in the motion differential equation. The beam element stiffness and mass matrices are assembled by matrix operations from the derived element field matrices. The usual finite element equations of motion for the rotor blade model are cast in the general case with the accounting for structural twist angle and possibly pre-bending.  So in the general case the rotor blade beam model represents a polygonal approximated space curve.

For the sake of verification, the natural frequencies and associated eigenmodes are calculated using real life rotor blade data with incorporation of realistic twist angle data. The first two edgewise and flapwise eigenfrequencies obtained are compared with the proposed results from FAST and ADAMS software given in (Jonkman, Butterfield, Musial, & Scott, 2009). It can be observed that the deviation of the results of Timoshenko beam model from FAST is comparatively lesser and is in good agreement with FAST and thus, it can be stated that the presented approach of alternative finite element formulation works

well.

One key input parameter for the Timoshenko beam model is the shear stiffness. As far it was not the main goal of the present work to determine an appropriate shear correction coefficient for realistic rotor blade data, the numerical example was performed with a very rough approximation for $GA_{sz}$ and $GA_{sy}$. It was used in order to simply demonstrate the performance and differences to the Bernoulli beam model. However, if detailed data for the complex multilayer design of a rotor blade are

available, more realistic estimation of the shear stiffness can be expected. A workable method for determination of the shear correction coefficient of a real life rotor blade represents an important topic for further research.

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
