# Peer review of "Determination of Natural Frequencies and Mode Shapes of a Wind Turbine Rotor Blades using Timoshenko Beam Elements"

_Wind Energy Science, 2018_

## Referee Comment (RC1) · Anonymous Referee #1 · 20 Nov 2018

The article by Stanoev and Chandrashekhara provides a valuable analysis of the determination of the free vibration properties of wind turbine blades.

The dynamic models considered, contains shear flexibility within the framework of Timoshenko beam theory. This is a suitable compromise for wind turbine blades, due to the large changes in cross section shape along the beam axis and the large cylindrical parts at the root. Thus, the model is appropriate for the analysis of modern wind turbine blades. A key feature of the numerical approach in the article, is the formulation of the beam stiffness via the static formulation using classic virtual work. The

formulation of the evolution format, integrated by a RK procedure, corresponds to the use of complementary virtual work, in which the static field comprise the virtual fields. Hereby, shear flexibility is included for free and variations in cross sectional properties are directly represented by the corresponding parameters appearing in the denominators of (13) or (14). The mathematic formulation appears to be elegant, although it is difficult to assess and verify all details in the formulation. The final results provide good agreement with the numerical results obtained by dedicated numerical packages. The paper is well suited for WES and should be accepted once the authors have addressed or commented on the following issues:

1. The analysis assumes an arbitrary location of the reference axis. This is very fine, as many simplified codes assumed decoupled kinematic effects. The authors still provide relations for a coinciding axis with the elastic center (see equations (11) and (14)). The paper would me more compact if the general case without the simplification by coinciding axes. Please consider whether it is necessary to include this particular case.

2. In (9)-(10) there is no coupling between torsion and shear. This implies that the chosen reference axis coincides with the shear center. Although this is done regularly in beam modelling, it is based on an assumption where coupling terms are neglected. The authors may note this in conjunction with equations (9)-(10).

3. In (13) and (14) many rows in the system matrices are zero rows, whereby the evolution of the section forces (strangely enough referred to as cutting forces in the paper) is explicitly attainable from the loading. Would it be possible to use the explicit relation to simply reduce the "redundant" section forces and thereby minimize the size of the system?

4. In the mass matrix (18) the structure is difficult to verify because zeros have not been included. Please consider to explicitly add the zero entries, as in (13)-(14).

5. It is stated that Hermitian interpolation is of 4th and 2nd order. As the polynomial

order is cubic I would think it would be interpolation of order 3 and 1. Consider to change the naming of order or be more specific of what you mean about e.g. 4th order interpolation.

6. The use of Runge-Kutta integration is elegant. Gauss integration is commonly used, as it is exact for polynomials, although the possess discontinuities at the element boundaries. For beam elements this is not required, whereby the alternative integration schemes is feasible. (This is just a comment.)

---

## Referee Comment (RC2) · Anonymous Referee #2 · 22 Nov 2018

I think this paper conveys the information, which is useful for the Wind Energy Science community. However, I've got an impression that the novelty and the significance of this submission are not clearly outlined.

The authors have included the paper by A. Bazoune, Y.A. Khulief and N.G. Stephen (JSV, 2003) in the list of references and mentioned it in Section 1 just in passing. However, it is not quite clear whether they have done anything novel in formulation of the stiffness matrix as compared with this reference. The formulation of the inertia matrix is reproduced from the earlier publication of the first author. I think the authors

should be more prudent in assessment of novelty of their contribution.

The numerical example, which aims to validate the proposed solution method is unconvincing. In Conclusions, lines 16.18, the authors acknowledge difficulties in assessment of shear correction coefficients and confess that the values used for their calculations (10% and 20%, page 15, line 9) are 'very rough approximation' (page 18, line 18). How sensitive are the frequencies reported in Table 1 to the choice of these parameters? What would give the Bernoulli-Euler model?

---

## Author Comment (AC1) · 23 Nov 2018

Evgueni Stanoev and Sudhanva Kusuma Chandrashekhara

evgueni.stanoev@uni-rostock.de

Thanks very much for your comments and remarks. My answers to your points:

1.The particular case for coinciding axis with the elastic centre is necessary to be included (and be developed to the final state) in order to apply the proposed numerical approach to a rotor blade model. The available rotor blade cross section data are with respect to principal elastic-centre axes. To apply the general case with arbitrary location of the beam reference axis, there will be needed all the stiffness values in the

matrix EA, see Eq.(8). Manufacturers of rotor blades do not release such data, same applies for shear stiffness Asy and Asz .

2. You are right, in this regular case the shear-torsion coupling terms are neglected by default. I will include a note after Eq. (9)-(10) to point it out explicitly.

3. It would be possible, but we are aiming to apply in an unified way the RK integration on an 1th order differential equation system from type eq.(38) – in order to achieve the regular form of the transfer matrix – see F(x,a)-matrix in eq.(40), with 12 state variables z1, z2. And, you are right, I change the "cutting" forces by "internal" or "section" forces at all places.

4. I will add all zero entries in Eq.(18).

5. I would change the notation of the Hermitian interpolation to 3th and 1th order. I mean the notation as e.g. 4th order interpolation points out that there are 4 unknown coefficients to be determined in a cubic polynomial.

---

## Referee Comment (RC3) · Anonymous Referee #1 · 26 Nov 2018

Thecomments and modifications provided by the author are satisfactory and thus I recommended publication of the submitted manuscript.

---

## Author Response (AR1)

**Determination of Natural Frequencies and Mode Shapes of a Wind Turbine Rotor Blades using Timoshenko Beam Elements**

Evgueni Stanoev[1], Sudhanva Kusuma Chandrashekhara[1],

[1]Faculty of Mechanical Engineering and Marine Technology, Endowed Chair of Wind Energy Technology, University of Rostock, Rostock, 18059, Germany

*Correspondence to*: PD Dr.-Ing. habil. Evgueni Stanoev (evgueni.stanoev@uni-rostock.de)

**Abstract.** In the simulation of a wind turbine, the lowest eigenmodes of the rotor blades are usually used to describe their elastic deformation in the frame of a multibody system. In this paper, a finite element beam model for the rotor blades based on the transfer matrix method is proposed. Both static and kinetic field matrices for the 3D Timoshenko beam element are derived by numerical integration of the differential equations of motion using RUNGE KUTTA 4[th] order procedure. The beam reference axis in the general case is at an arbitrary location in the cross section. The inertia term in the motion differential equation is expressed using appropriate shape functions for the Timoshenko beam. The kinetic field matrix is built by numerical integration applied on the approximated inertia term. The beam element stiffness and mass matrices are calculated by simple matrix operations from both field matrices. The system stiffness and mass matrices of the rotor blade model are assembled in the usual finite element manner in a global coordinate system with the accounting for structural twist angle and possibly pre-bending. The program developed for the above calculations and the final solution of the eigenvalue problem is accomplished using MuPAD, a symbolic math toolbox of MATLAB[®]. The calculated natural frequencies using generic rotor blade data are compared with the results proposed from FAST and ADAMS software.

**1   Introduction**

Vibration of an elastic system refers to a limited reciprocating motion of a particle or an object of the system. Wind turbines operate in an unsteady environment which results in a vibrating response (Manwell, McGowan, & Rogers, 2009). They consist of long slender structures (rotor blades and tower), of which resonant frequencies should be taken into account during the initial design and construction. When the excitation frequency of the vibrating system is near any natural frequency, the undesirable resonant state occurs with large amplitudes, which may lead to damage or even collapse of the wind turbine or its components. The vibration response especially of the rotor blades depends on the stiffness which is a function of the materials used, design and size ( Jureczko, Pawlak, & M̦ e˙zyk, 2005). Therefore, the estimation of natural frequencies in the early design phase plays an important role in avoiding resonance.

The eigenmodes associated to the lowest natural frequencies are employed as shape functions to describe the elastic deformation of the rotor blade beam model in the frame of the usual simulation of the wind turbine as a multi-body system.

Mostly, the first two bending eigenmodes in each (flapwise and edgewise) direction and optionally the two additional torsional eigenmodes are used. The determination of those lowest eigenmodes with sufficient numerical accuracy is the first step for the modal superposition applied to the deformational motion of the rotor blades.

Due to geometrical complexity of the blade cross section profiles and the extended use of composite materials, the exact
5    calculation of natural frequencies in the design process takes a considerable time and financial expense for the 3D modelling of the rotor blade using CAD software. Hence a simplified finite element beam model is necessary. A twisted rotor blade is simplified into a cantilever beam with non-uniform cross section. The structural twist angle is implemented by changing the orientation of the principal axis of the blade cross section along the length of the blade.

In the finite element formulation of beams two linear beam theories are established, the Euler Bernoulli beam model and the
10   Timoshenko beam model. Although the Euler-Bernoulli beam theory is widely used, the Timoshenko beam theory is considered to be better as it incorporates the effects of transverse shear and the rotational inertia on the dynamic response of the beam (Kaya, 2006). In the classical approach of finite element formulation for the free vibration analysis of beams, the stiffness and mass matrices are derived using interpolation functions derived from second and fourth order Hermite polynomials. The stiffness matrix is derived from the equation (Wu, 2013):

15   $$\underline{\underline{K}}(e) = \int \underline{\underline{B}}^T \underline{\underline{D}}_m \, \underline{\underline{B}} \, dv \qquad\qquad (1)$$

where, $\underline{\underline{K}}(e)$ is the element stiffness matrix, $\underline{\underline{B}}$ is the strain matrix , $\underline{\underline{D}}_m$ is the elasticity matrix for the beam. The element mass matrix of the beam is derived using the equation (Wu, 2013):

$$\underline{\underline{M}}(e) = \int \rho \, \underline{\underline{a}}^T \underline{\underline{a}} \, dv \qquad\qquad (2)$$

[revised manuscript text omitted]

$$
\begin{bmatrix}
H_{u_{\overline{1}}} & — & — & — & — & — & H_{u_{\overline{2}}} & — & — & — & — & — \\
— & H_{v_{\overline{1}}} & — & — & — & H_{v_{\overline{2}}} & — & H_{v_{\overline{3}}} & — & — & — & H_{v_{\overline{4}}} \\
— & — & H_{w_{\overline{1}}} & — & H_{w_{\overline{2}}} & — & — & — & H_{w_{\overline{3}}} & — & H_{w_{\overline{4}}} & — \\
— & — & — & H_{u_{\overline{1}}} & — & — & — & — & — & H_{u_{\overline{2}}} & — & — \\
— & — & H_{\varphi_{y_{\overline{1}}}} & — & H_{\varphi_{y_{\overline{2}}}} & — & — & — & H_{\varphi_{y_{\overline{3}}}} & — & H_{\varphi_{y_{\overline{4}}}} & — \\
— & H_{\varphi_{z_{\overline{1}}}} & — & — & H_{\varphi_{z_{\overline{2}}}} & — & H_{\varphi_{z_{\overline{3}}}} & — & — & — & H_{\varphi_{z_{\overline{4}}}}
\end{bmatrix}

[revised manuscript text omitted]

**Anonymous Referee #1

The article by Stanoev and Chandrashekhara provides a valuable analysis of the determination of the free vibration properties of wind turbine blades. The dynamic models considered, contains shear flexibility within the framework of Timoshenko beam theory. This is a suitable compromise for wind turbine blades, due to the large changes in cross section shape along the beam axis and the large cylindrical parts at the root. Thus, the model is appropriate for the analysis of modern wind turbine blades. A key feature of the numerical approach in the article, is the formulation of the beam stiffness via the static formulation using classic virtual work. The C1 WESD Interactive comment Printer-friendly version Discussion paper formulation of the evolution format, integrated by a RK procedure, corresponds to the use of complementary virtual work, in which the static field comprise the virtual fields. Hereby, shear flexibility is included for free and variations in cross sectional properties are directly represented by the corresponding parameters appearing in the denominators of (13) or (14). The mathematic formulation appears to be elegant, although it is difficult to assess and verify all details in the formulation. The final results provide good agreement with the numerical results obtained by dedicated numerical packages. The paper is well suited for WES and should be accepted once the authors have addressed or commented on the following issues:

1. The analysis assumes an arbitrary location of the reference axis. This is very fine, as many simplified codes assumed decoupled kinematic effects. The authors still provide relations for a coinciding axis with the elastic center (see equations (11) and (14)). The paper would me more compact if the general case without the simplification by coinciding axes. Please consider whether it is necessary to include this particular case.

Answer: The particular case for coinciding axis with the elastic centre is necessary to be included (and be developed to the final state) in order to apply the proposed numerical approach to a rotor blade model. The available rotor blade cross section data are with respect to principal elastic-centre axes. To apply the general case with arbitrary location of the beam reference axis, there will be needed all the stiffness values in the matrix

$$\begin{bmatrix} EA & EA_y & EA_z \\ EA_y & EA_{yy} & EA_{yz} \\ EA_z & EA_{yz} & EA_{zz} \end{bmatrix}, \text{ see Eq.(8). Manufacturers of rotor blades do not release such data,}$$

same applies for shear stiffness $A_{sz}$ and $A_{sy}$.

No changes have been made.

2. In (9)-(10) there is no coupling between torsion and shear. This implies that the chosen reference axis coincides with the shear center. Although this is done regularly in beam modelling, it is based on an assumption where coupling terms are neglected. The authors may note this in conjunction with equations (9)-(10).

Answer: You are right, in this regular case the shear-torsion coupling terms are neglected by default. I will include a note after Eq. (9)-(10) to point it out explicitly.

Changes in manuscript: After Eq.(10):

The relation (10a,b) implies that the chosen reference axis coincides with the shear centre – due to neglected shear-torsion coupling terms in Eq.(7).

3. In (13) and (14) many rows in the system matrices are zero rows, whereby the evolution of the section forces (strangely enough referred to as cutting forces in the paper) is explicitly attainable from the loading. Would it be possible to use the explicit relation to simply reduce the "redundant" section forces and thereby minimize the size of the system?

Answer: It would be possible, but we are aiming to apply in an unified way the RK integration on an 1th order differential equation system from type eq.(38) – in order to achieve the regular form of the transfer matrix – see $\underline{\underline{F}}(x, a)$-matrix in eq.(40), with 12 state variables $\begin{bmatrix} \underline{z}_1(a) \\ \underline{z}_2(a) \end{bmatrix}$. And, you are right, I change the "cutting" forces by "internal" or "section" forces at all places.

Changes: Replacing the "cutting" forces by "internal" or "section" forces at all places.

4. In the mass matrix (18) the structure is difficult to verify because zeros have not been included. Please consider to explicitly add the zero entries, as in (13)-(14).

Answer: I will add all zero entries in Eq.(18).

Changes:

$$\underline{\underline{m}} = \mu \cdot \begin{bmatrix} 1 & 0 & 0 & 0 & z & -y \\ 0 & 1 & 0 & -z & 0 & 0 \\ 0 & 0 & 1 & y & 0 & 0 \\ 0 & -z & y & \left(y^2 + z^2 + \frac{\Theta_p}{\mu}\right) & 0 & 0 \\ z & 0 & 0 & 0 & \left(z^2 + \frac{\Theta_y}{\mu}\right) & -yz \\ -y & 0 & 0 & 0 & -yz & \left(y^2 + \frac{\Theta_z}{\mu}\right) \end{bmatrix} \tag{18}$$

5. It is stated that Hermitian interpolation is of 4th and 2nd order. As the polynomial order is cubic I would think it would be interpolation of order 3 and 1. Consider to change the naming of order or be more specific of what you mean about e.g. 4th order interpolation.

Answer: I would change the notation of the Hermitian interpolation to 3th and 1th order. I mean the notation as e.g. $4^{th}$ order interpolation points out that there are 4 unknown coefficients to be determined in a cubic polynomial.

Changes: In Sec. 3, page 7:

In the classical finite element formulation, the beam stiffness matrix and the consistent beam mass matrix are derived by developing an approach for the displacement functions through shape (interpolation) functions, which consists of $1^{nd}$ and $3^{th}$ order Hermite polynomials.

6. The use of Runge-Kutta integration is elegant. Gauss integration is commonly used, as it is exact for polynomials, although the possess discontinuities at the element boundaries. For beam elements this is not required, whereby the alternative integration schemes is feasible. (This is just a comment.)

No answer, no comment.

**Anonymous Referee #2

1) I think this paper conveys the information, which is useful for the Wind Energy Science community. However, I've got an impression that the novelty and the significance of this submission are not clearly outlined. The authors have included the paper by A. Bazoune, Y.A. Khulief and N.G. Stephen (JSV, 2003) in the list of references and mentioned it in Section 1 just in passing. However, it is not quite clear whether they have done anything novel in formulation of the stiffness matrix as compared with this reference.

Answer: In connection with your first remark to the cited paper by Bazoune, A., & Khulief, Y. (2003) : we have used just the shape functions for Timoshenko beam elements, presented in (JSV,2003), and have described them detailed in the article – Eqs. (31), (32), (35), (36). They are employed, see Eq. (37), only for the inertia-forces term in Eq.(17), and have not been used in the derivation of the element stiffness matrix K(e) in Eq. (43). The stiffness matrix is developed by numerical Runge-Kutta integration of the coefficient matrix A of the differential equations system, Eq.(16), for Timoshenko beam – in the form of Eq. (14). In the frame of this integration, described in Sec. 3.4, first the static field matrix L(e), see Eq. (41), is built, and finally the element stiffness matrix K(e) - by the operations shown in Eq.(43).

No changes.

2) The formulation of the inertia matrix is reproduced from the earlier publication of the first author. I think the authors should be more prudent in assessment of novelty of their contribution.

Answer: The formulation of the element mass matrix by similar numerical integration of the inertia-force matrix $b_m$, see Eq.(17)-(18), is in fact reproduced from earlier publication of the first author (Stanoev 2007) where only Bernoulli beam element is considering. In the present article are combined the numerical procedure for assembly of the element mass matrix with the above mentioned Timoshenko shape functions.

3) The numerical example, which aims to validate the proposed solution method is unconvincing. In Conclusions, lines 16.18, the authors acknowledge difficulties in assessment of shear correction coefficients and confess that the values used for their calculations (10% and 20%, page 15, line 9) are 'very rough approximation' (page 18, line 18). How sensitive are the frequencies reported in Table 1 to the choice of these parameters? What would give the Bernoulli-Euler model?

Answer: To your remarks on the numerical example, I admit, we haven't study the sensitivity of the eigenfrequencies to the choice of the shear correction coefficients. I will add, similarly to Tab.1, two additional tables with different shear coefficients approximation and for comparison of Bernoulli to Timoshenko beam model.

Changes: on page 18:

| $\dfrac{GA_{sz}}{EA}$ : $\dfrac{GA_{sy}}{EA}$

 **Eigenmode Type** | **Timoshenko Beam** | | | **Bernoulli Beam** |
|---|---|---|---|---|
| | **10% : 20%**

 $f_{\text{Tim}}$ $[Hz]$ | **20% : 40%**

 $f_{\text{Tim}}$ $[Hz]$ | **30% : 60%**

 $f_{\text{Tim}}$ $[Hz]$ |

 $f_{\text{Bern}}$ $[Hz]$ |
| 1$^{st}$ Flapwise Bending Mode | 0.6704 | 0.6737 | 0.6749 | 0.6771 |
| 1$^{st}$ Edgewise Bending Mode | 1.0958 | 1.1035 | 1.1060 | 1.1113 |
| 2$^{nd}$ Flapwise Bending Mode | 1.8992 | 1.9227 | 1.9307 | 1.9472 |
| 1$^{st}$ Mixed Flap/Edge Mode | 3.8357 | 3.9275 | 3.9596 | 4.0262 |
| 2$^{nd}$ Mixed Flap/Edge Mode | 4.2922 | 4.4062 | 4.4462 | 4.5295 |
| 1$^{st}$ Torsional Mode | 5.5181 | 5.5181 | 5.5181 | 5.5181 |
| 2$^{nd}$ Torsional Mode | 9.6937 | 9.6937 | 9.6937 | 9.6937 |

**Table 2: Comparison Timoshenko – Bernoulli beam with 3 variants for shear stiffness values**

In table 2 are shown the calculated natural frequencies for three different variants for the shear correction coefficients, approximated as $\frac{GA_{sz}}{EA}$ resp. $\frac{GA_{sy}}{EA}$ –ratios. The comparison to the frequencies calculated using the Bernoulli-beam model outlines the tendency to more stiff structure due to the presupposed infinite shear stiffness in this case. Natural frequencies $f_{\text{Bern}}$ are on average 0.5% -1.0% higher then $f_{\text{Tim}}$ - in the (30% : 60%)-case. The natural frequencies remain unchanged for both beam models only for the purely torsional modes. The reason is that the equations for torsion and bending are uncoupled (for the case of principal axes, see Eq. (14)) and remain the same in both models.